# Diabetes-Related Changes in Carotid Wall Properties: Role of Triglycerides

**DOI:** 10.3390/jcm13185654

**Published:** 2024-09-23

**Authors:** Michaela Kozakova, Carmela Morizzo, Giuseppe Penno, Dante Chiappino, Carlo Palombo

**Affiliations:** 1Department of Clinical and Experimental Medicine, University of Pisa, 56126 Pisa, Italy; michaela.kozakova@esaote.com (M.K.); giuseppe.penno@unipi.it (G.P.); 2Esaote SpA, 16153 Genova, Italy; 3School of Medicine, Department of Surgical, Medical, Molecular Pathology and Critical Care Medicine, University of Pisa, 56126 Pisa, Italy; carmela.morizzo@unipi.it; 4Fondazione Toscana G. Monasterio, Massa-Pisa, 56124 Pisa, Italy; dchiappino@gmail.com; 5Department of Surgical, Medical, Molecular Pathology and Critical Care Medicine, University of Pisa, Via Savi 10, 56126 Pisa, Italy

**Keywords:** diabetes mellitus, carotid artery, arterial stiffness, triglycerides, radiofrequency signal

## Abstract

**Background/Objectives:** This study compares the power of the radiofrequency (RF) signal reflected from the media layer (media power) of the common carotid artery (CCA) and the CCA stiffness between individuals with and without type 2 diabetes mellitus (T2DM). It also evaluates the associations of CCA media power with plasma glucose and lipid levels, as well as carotid stiffness. **Methods:** A total of 540 individuals, 115 with and 425 without T2DM (273 males, mean age = 64 ± 8 years) were studied using RF-based tracking of the right CCA. The following parameters were measured: CCA media thickness, luminal diameter, wall tensile stress (WTS), local pulse wave velocity (PWV), and media power. **Results:** Compared to the non-diabetic individuals, the T2DM patients had significantly higher CCA media thickness (652 ± 122 vs. 721 ± 138 microns, *p* < 0.005), luminal diameter (6.12 ± 0.78 vs. 6.86 ± 0.96 mm, *p* < 0.0005), media power (36.1 ± 4.8 vs. 39.3 ± 4.6, *p* < 0.0001), and PWV (7.65 ± 1.32 vs. 8.40 ± 1.89 m/s; *p* < 0.01), but comparable WTS (32.7 ± 10.4 vs. 33.1 ± 10.7 kPa; *p* = 0.25). In the entire population, CCA media power was independently associated with male sex, pulse pressure, current smoking, and T2DM; when T2DM was not included in the model, triglycerides emerged as an independent determinant of media power. The CCA PWV was independently associated with age, pulse pressure, media power, and T2DM. **Conclusions:** Our findings suggest the presence of structural changes in the arterial media of T2DM patients, leading to carotid stiffening and remodeling, aiming to preserve WTS. T2DM-related changes in arterial wall composition may be driven by high plasma triglyceride levels, which have previously been associated with both arterial stiffening and the incidence of CV events.

## 1. Introduction

The incidence of cardiovascular (CV) disease among individuals with type 2 diabetes mellitus (T2DM) is 2–3 times higher than among individuals without diabetes. CV risk factors such as obesity, high blood pressure (BP), and dyslipidemia are common in T2DM patients, placing them in the high-risk category; individuals with diabetes enter the high CV disease risk category (a 10-year risk of 20% or more) 15 years earlier than non-diabetic individuals [1,2,3].

Arterial stiffness is a strong predictor of future CV events and mortality [4], and T2DM patients experience an accelerated age-related decline in arterial compliance [5,6,7]. Previous studies have shown that the impact of T2DM is greater in the heart–carotid and heart–femoral segments of the arterial tree (large elastic arteries) than in the heart–brachial and femoral–ankle segments (medium-sized muscular arteries) [8,9]. As a result, T2DM patients exhibit greater aortic and carotid stiffness at any given level of systolic blood pressure (BP) [5,6,7,10,11], and the aortic stiffness of T2DM patients is comparable to that of non-diabetic individuals who are 15 years older [9].

The elasticity and integrity of the arterial wall are maintained by the extracellular matrix, particularly elastin and collagen, and by vascular smooth muscle cells (VSMCs). Elastin fragmentation, increased collagen synthesis and cross-linking, and VSMC dedifferentiation are all involved in arterial wall stiffening [12,13,14]. T2DM may accelerate arterial stiffening by several mechanisms, including the formation and accumulation of advanced glycation end-products (AGEs), endothelial dysfunction, inflammation, and dyslipidemia [15,16,17,18,19]. AGEs contribute to vascular stiffening not only by the formation of cross-links on long-lived proteins such as collagen, but also by reducing nitric oxide bioavailability [20], increasing reactive oxygen species (ROS) formation [16], and stimulating endothelin-1 transcription in endothelial cells [21]. An interesting study using ^18^F-fluorodeoxyglucose positron emission tomography (FDG-PET) revealed a significant impact of T2DM on FDG uptake in the carotid wall, with the degree of uptake increasing with fasting glucose levels [18]. Other studies have also suggested the association between arterial stiffness and plasma lipid levels [19,22,23]. Therefore, targeting the factors contributing to arterial stiffening in patients with T2DM could help to reduce diabetes-related CV morbidity.

Structural alterations in the arterial wall can be detected by ultrasound (US), as the mechanical energy of the propagating US interacts with the material of the arterial media. The information from this ultrasound–tissue interaction is contained in the reflected signal, which is captured by the US transducer and converted into an electrical signal known as a radiofrequency (RF) signal [24]. Arterial wall changes can be assessed either through the densitometric analysis of B-mode images or by analyzing the RF signal. In previous studies, a first-order densitometric analysis of the B-mode carotid images showed a direct correlation between the mean gray level of the carotid plaque shoulder and the content of VSMCs, as assessed by immunocytochemistry (r = 0.58) [25], and the integrated backscatter power of the carotid media was correlated directly with both the elastic fragmentation index and the collagen fiber index, as determined by histological examination [26].

In the present study, we compared the power of the signal reflected from the media layer (media power) of the common carotid artery (CCA), as well as the CCA geometry and stiffness, between T2DM patients and non-diabetic individuals free of CV events. We also assessed the association of CCA media power with body size, BP, plasma glucose and lipid levels, and carotid stiffness.

## 2. Materials and Methods

### 2.1. Study Population and Protocol

The study population consisted of 540 individuals without a history of myocardial ischemia (symptoms, ECG), myocardial infarction and percutaneous coronary procedures, transient ischemic attack and ictus, peripheral artery disease, or carotid plaque in the CCA. The participants were voluntarily recruited from the prospective cohort study, “MHeLP, Montignoso Heart and Lung Project”, during follow-up visits between December 2015 and January 2022. All participants underwent an examination protocol that included medical history, anthropometry, brachial BP measurements, fasting blood test, ECG, and a high-resolution carotid ultrasound. Diabetes mellitus was defined as fasting glucose ≥ 7.0 mmol/L or 2 h plasma glucose ≥ 11.1 mmol/L confirmed by a second test, or treatment for diabetes [27]. Type 1 diabetes mellitus was ruled out based on medical history, insulin, and C-peptide plasma levels. Hypertension was defined as systolic BP  > 140 mmHg and/or diastolic BP  > 90 mmHg [28].

The study protocol adhered to the ethical guidelines of the 1975 Declaration of Helsinki, and was approved by the “Comitato Etico di Area Vasta Nord-Ovest” (CEAVNO), approval code number 2514, approval date 1 September 2008, at the beginning of the MHeLP study. All individuals gave their informed consent to participate.

### 2.2. Body Size and BP Measurement

Body weight (kg) and height (m) were measured, and body mass index (BMI, kg/m^2^) was calculated. Waist circumference (cm) was measured as the narrowest circumference between the lower rib margin and the anterior superior iliac crest. BP was measured at two different visits by a validated digital electronic tensiometer (Omron, model 705cp, Kyoto, Japan). Measurements were taken with participants seated for at least 10 min, using regular or large adult cuffs, depending on arm circumference. Two measurements were taken at each visit, and the average was calculated. The mean of the two visits was used to estimate brachial BP (mmHg). Brachial pulse pressure was calculated as the difference between systolic and diastolic BP.

### 2.3. CCA Intima–Media Thickness, Luminal Diameter, Wall Tensile Stress, Local Pulse Wave Velocity, Media Thickness and Media Power

Carotid ultrasound was performed on the right CCA by a single operator blinded to the diagnosis of the participants, using an ultrasound scanner equipped with a 10 MHz linear probe (MyLabOne, Esaote, Genova, Italy) and RF-based tracking of the arterial wall (QIMT^®^, QAS^®^), which automatically determines far-wall intima–media thickness (IMT), inter-adventitial diameter (IAD) and distension with high spatial and temporal resolution (sampling rate of 550 Hz on 32 lines) [29,30,31]. CCA structure and function were assessed within a rectangular 1 cm long ROI placed 1 cm before the flow divider. All participants were asked to abstain from cigarette smoking, caffeine and alcohol consumption, and vigorous physical activity for 24 h prior to the examination.

CCA IMT was defined as the distance between the lumen–intima and the media–adventitia interfaces of the CCA far (posterior) wall at diastole, and IAD as the distance between the media–adventitia interfaces of the near and far wall at diastole. CCA luminal diameter (mm) was calculated as IAD-(2*IMT) [32]. Carotid wall tensile stress (WTS; kPa) was calculated according to Laplace’s law as pulse pressure*(r/w), where r is the luminal radius (luminal diameter/2) and w is the wall thickness (far-wall IMT) [33]. Local carotid pulse wave velocity (PWV; m/s) was estimated from distension curves using the Bramwell–Hill equation, which relates propagation velocity to arterial distensibility through the following equation: CCA PWV = √ ρ × DC, where ρ is the blood density and DC is distensibility coefficient describing the absolute change in vessel diameter (Δ*D*) during cardiac cycle for a given change in local pressure (Δ*p*) [31,34]. The local carotid pressure used for PWV calculation was estimated by the QAS system, converting the distension curve to pressure curve by a linear conversion factor and assuming that the difference between mean arterial pressure and diastolic pressure is invariant along the arterial tree [35]. The peripheral BP needed for rescaling was measured at the left brachial artery (Omron, Kyoto, Japan) during each acquisition of the distension curves.

The ultrasound system was modified to allow the raw RF signal to be transmitted and stored on a personal computer and later analyzed using MATLAB programming platform (MathWorks, Natick, MA, USA). Peak signal-to-noise ratio (PSNR) of media layer was calculated in the time domain. PSNR, or media power, represents the ratio between the maximum signal power (*p*) and the signal noise power, defined as variance (var) and calculated in the center of the vessel. Media power was expressed as 10 log(P/var).

RF-derived measures (IMT and distension) represent the average of 6 consecutive cardiac beats. The mean of two acquisitions was used for statistical analysis. Intra- and interindividual variability was assessed in 25 volunteers, with acquisitions performed in two separate sessions 30 min apart, both by the same operator and by two different operators. Brachial pulse pressure was consistent across different acquisitions (*p* = 0.88). Intra- and interindividual variability for IMT and distension measurements were 6.7 ± 4.2 and 8.7 ± 6.4%, and 7.5 ± 4.6 and 9.0 ± 6.9%, respectively.

### 2.4. Statistical Analysis

Data are expressed as mean ± SD, categorical data as percentages. Variables with skewed distribution were summarized as median (interquartile range), and were logarithmically transformed for parametric statistical analysis. ANOVA was used to compare continuous variables, while the χ^2^ test was applied for categorical variables. The association of CCA media power and PWV with variables related to T2DM such as body size and plasma levels of lipids and glucose were assessed by univariate regression analysis. To identify the independent determinants of CCA media power and PWV, multiple regression analysis with backward stepwise removal was performed.

Two multiple regression analyses were conducted; in the first, we tested the independent associations of vascular measures with sex, age, BP, current smoking, hypertensive and lipid-lowering therapy, and the presence of T2DM. In the second analysis, we examined independent associations with sex, age, BP, current smoking, hypertensive and lipid-lowering therapy, and T2DM-related variables that were significantly associated with vascular measures in the univariate analysis (*p* < 0.05). Statistical tests were two-sided, and significance was set at a value of *p* < 0.05. Statistical analysis was performed by JMP software, version 3.1 (SAS Institute Inc., Cary, NC, USA).

## 3. Results

Characteristics of the study population are reported in Table 1, and the differences in established CV risk factors between individuals with and without T2DM in Table 2. The T2DM patients were more often men, were older, and had higher body size, plasma levels of triglycerides (TGs), and fasting glucose, and lower levels of HDL and LDL cholesterol. They also had a higher prevalence of hypertensive and lipid-lowering treatment. Brachial BP and current smoking did not differ between the two groups.

Table 3 compares the CCA geometry and function between individuals with and without T2DM, after adjustment for sex, age, and BMI. The diabetic patients had significantly higher CCA IMT, media thickness, luminal diameter, media power, and PWV compared to the non-diabetic individuals. The WTS was comparable between the two groups.

In the entire population, the CCA media power increased with waist circumference, pulse pressure, TGs, and fasting glucose (r = 0.15–0.23; *p* < 0.005–0.0001), and decreased with HDL cholesterol (r = −0.22; *p* < 0.0001). The CCA PWV increased with media power (r = 0.29; *p* < 0.0001), as well as with age, waist circumference, pulse pressure, TGs, and fasting glucose (r = 0.14–0.40; *p* < 0.005–0.0001), and decreased with HDL cholesterol (r = −0.14; *p* < 0.005). In addition, the plasma TGs increased and HDL cholesterol decreased with fasting plasma glucose levels (r = 0.25 and −0.29, respectively; *p* < 0.0001 for both).

Table 4 demonstrates the results of the multiple regression analyses for CCA media power and CCA PWV across the entire population. In the first analysis, which included T2DM as the independent variable, media power was independently associated with male sex, pulse pressure, current smoking, and T2DM. In the second analysis, which included T2DM-related variables, media power was independently associated with male sex, pulse pressure, current smoking, and TG levels. CCA PWV was independently associated with age, pulse pressure, media power, and T2DM in the first analysis, and with age, pulse pressure, media power, and fasting glucose levels in the second analysis.

## 4. Discussion

In this study, patients with T2DM exhibited significantly higher CCA media power and PWV compared to non-diabetic individuals. The increase in media power likely reflects alterations in the extracellular matrix of the arterial wall, a hypothesis supported by both experimental and clinical studies. In an “ex vivo” study, the integrated backscatter signal from freshly excised human aortic segments increased from normal to fibrous and calcified regions [36]. In an experimental study on the canine ascending aorta, the backscatter coefficient for elastin-isolated tissue was found to be five times higher than that of collagen-isolated tissue, suggesting that elastin fibers are the primary scattering components within elastic arteries [37]. In a human study, the integrated backscatter value of the carotid media obtained ante mortem was correlated with both the elastic fragmentation index and the collagen fiber index (r = 0.63 and 0.59, respectively) in histological specimens. Additionally, the integrated backscatter was correlated directly with the carotid beta stiffness index [26].

T2DM can induce changes in arterial wall composition through various mechanisms related to its metabolic dysregulation and systemic inflammation [15,16,17,18,19,20,21,22,23]. In the present study, T2DM was identified as an independent determinant of media power. However, when T2DM-related variables, like body size and plasma lipid and glucose levels, were included in the model, TG levels emerged as independent determinants of media power, which, in turn, was an independent determinant of carotid stiffness. 

Hypertriglyceridemia is the most common lipid abnormality in T2DM, and several studies have demonstrated a link between TGs and arterial stiffness [19,22,38,39], as well as between TGs and CV events [22,40]. In healthy men, TG levels were found to be associated with augmentation index independently of other cardiometabolic risk factors [38]. Similarly, in a community-based population in China, plasma TG levels were independently associated with both carotid–femoral and carotid–radial PWV, and changes in TG levels over a 4.8-year period were correlated with changes in carotid–femoral PWV [39]. The impact of TGs on CV events was highlighted in a Danish study of newly diagnosed T2DM patients without previous CV disease, in which TG levels were associated with major adverse cardiac events, starting at a level of 1.0 mmol/L [40]. Moreover, a recent meta-analysis of randomized controlled trials showed that TG-lowering therapy in diabetic patients resulted in a reduced risk of CV events (RR = 0.91, 95% CI 0.87–0.95), independent of the degree of TGs reduction and glycemic control [41].

The mechanisms linking TGs to the structural changes in media and arterial stiffening are not fully understood, but experimental studies offer some potential explanations. TGs are composed of three fatty acids esterified to a glycerol molecule, and elastin has a propensity to associate with fatty acids. The resulting elastin–fatty acid complexes are more susceptible to elastolysis than elastin itself [42,43,44]. Certain saturated fatty acids, such as palmitic acid, may also promote medial calcification by enhancing the production of reactive oxygen species, which stimulate extracellular-signal-regulated kinase (ERK1/2)-mediated osteogenic gene expression and osteogenic differentiation of VSMCs [45]. In warfarin/vitamin K-treated Wistar rats, the addition of palmitic acid to the diet increased aortic calcification by 2.4-fold, and this increase was associated with a significant rise in aortic PWV [45]. Taken together, these findings suggest that TGs may accelerate elastin fragmentation in the carotid media and induce medial calcification, both of which could enhance the power of reflected signals and increase carotid stiffness. As expected, carotid stiffness also increased with increasing fasting plasma glucose levels [46]. 

Other independent determinants of media power, aside from T2DM and TGs, include pulse pressure, current smoking, and male sex. Chronic exposure to high pulsatile load exerts a fatiguing effect on the load-bearing elements of the arterial media, mainly on elastin, causing its fracture and fragmentation [13]. Exposure to tobacco smoke has been shown to increase the number of elastin breaks in the thoracic and abdominal aorta of mice [47], as well as to elevate the content of VSMCs and the extracellular matrix in the aortic wall [48]. The impact of sex on the echo-reflectivity of the carotid media is less clear, but it may be explained by the influence of sex hormones on VSMC proliferation and migration, as well as the sex-specific expression of mineralocorticoid receptors (MRs) in VSMCs. Endogenous estrogen and progesterone inhibit VSMC proliferation and migration [49,50,51,52], while the MRs in VSMCs may accelerate age-related vascular fibrosis and stiffening, particularly in males [53]. 

Structural changes in the carotid media can induce arterial remodeling, as the loss of media elastic function leads to luminal enlargement and increases in WTS [54]. Increased tensile stress activates intracellular signaling pathways, which promotes VSMCs proliferation and migration within the media [55], resulting in wall thickening and subsequent stress reduction. Indeed, our T2DM patients exhibited higher CCA PWV and luminal diameter, but comparable WTS due to higher media thickness.

### Study Limitations

The T2DM patients and non-diabetic individuals were not comparable for sex, age or body size, all of which contribute to arterial stiffening and remodeling. However, analyses comparing the vascular measures in diabetic and non-diabetic individuals were adjusted for sex, age and BMI. Markers of inflammation, which might contribute to structural changes in the carotid media of T2DM patients, were not assessed.

## 5. Conclusions

This is the first study to analyze the power of the signal reflected from the carotid wall, with the aim of obtaining information on diabetes-related alterations in the carotid media. Our findings indicate that hypertriglyceridemia, the most common diabetic lipid abnormality, may trigger structural changes in arterial media, leading to arterial stiffening and remodeling. Since arterial stiffness is a strong predictor of CV mortality, and since the T2DM patients had accelerated arterial stiffening, strict TG control could reduce CV risk in diabetic patients.

## Figures and Tables

**Table 1 jcm-13-05654-t001:** Characteristics of Study Population.

	Mean ± SD/Median (IQR)/n (%)	Range
Sex—Male:Female	273 (49):267 (51)	
Age (years)	64 ± 8	41–90
BMI (kg/m^2^)	27.2 ± 4.1	15.5–51.7
Waist circumference (cm)	96 ± 12	64–139
Brachial systolic BP (mmHg)	134 ± 20	96–198
Brachial pulse pressure (mmHg)	57 ± 16	25–105
HDL cholesterol (mmo/L)	1.53 ± 0.42	0.44–3.05
LDL cholesterol (mmo/L)	3.15 ± 0.88	0.91–6.43
TGs (mmo/L)	1.06 [0.77]	0.23–4.28
Fasting glucose (mmol/L)	5.70 ± 1.27	2.39–13.39
Current smoker (yes)	94 (17)	
Hypertension therapy (yes)	130 (24)	
T2DM (yes)	115 (21)	
Lipid-lowering therapy (yes)	126 (23)	

**Table 2 jcm-13-05654-t002:** Established Cardiovascular Risk Factors in Individuals with and without T2DM.

	Mean ± SD/Median (IQR)/n (%)		*p* *
	T2DM (No)	T2DM (Yes)	
	425	115	
Sex (male)	189 (45)	84 (73)	<0.0005
Age (years)	63 ± 8	67 ± 8	<0.0001
BMI (kg/m^2^)	26.8 ± 4.1	28.7 ± 4.0	<0.0005
Waist circumference (cm)	95 ± 12	103 ± 12	<0.0001
Brachial systolic BP (mmHg)	134 ± 20	132 ± 18	0.09
Brachial pulse pressure (mmHg)	57 ± 17	57 ± 15	0.14
HDL cholesterol (mmo/L)	1.60 ± 0.42	1.28 ± 0.34	<0.0001
LDL cholesterol (mmo/L)	3.28 ± 0.83	2.69 ± 0.92	<0.0001
TGs (mmo/L)	0.98 [0.75]	1.23 [0.90]	0.0001
Fasting glucose (mmol/L)	5.27 ± 0.62	7.35 ± 1.70	<0.0001
HbA1c (%)		45.4 [13.2]	
Current smoker (yes)	70 (17)	24 (21)	0.32
Hypertension therapy (yes)	81 (19)	49 (43)	<0.0001
Lipid-lowering therapy (yes)	64 (15)	62 (54)	<0.0001

*: adjusted for sex and age.

**Table 3 jcm-13-05654-t003:** CCA Geometry and Function in Individuals with and without T2DM.

	Mean ± SD		*p* *
	T2DM (No)	T2DM (Yes)	
Luminal diameter (mm)	6.12 ± 0.78	6.86 ± 0.96	<0.0005
IMT (microns)	725 ± 135	802 ± 153	<0.005
Media thickness (microns)	652 ± 122	721 ± 138	<0.005
Wall tensile stress (kPa)	32.7 ± 10.4	33.1 ± 10.7	0.25
Media power	36.1 ± 4.8	39.3 ± 4.6	<0.0001
PWV (m/s)	7.65 ± 1.32	8.40 ± 1.89	<0.01

*: adjusted for sex, age and BMI.

**Table 4 jcm-13-05654-t004:** Independent Determinants of CCA Media Power and PWV.

		Model with T2DM		Model with T2DM-Related Factors	
		Beta ± SE	*p*	Beta ± SE	*p*
**CCA Media power**	Sex (male)	0.28 ± 0.04	<0.0001	0.31 ± 0.04	<0.0001
	PP (mmHg)	0.15 ± 0.04	<0.0005	0.13 ± 0.04	<0.005
	Smoking (yes)	0.14 ± 0.05	<0.01	0.12 ± 0.05	<0.05
	T2DM (yes)	0.24 ± 0.04	<0.0001		
	logTGs			0.14 ± 0.04	0.001
	Cumulative R^2^	0.18	<0.0001	0.16	<0.0001
**CCA PWV (m/s)**	Age (years)	0.13 ± 0.04	<0.005	0.13 ± 0.04	<0.005
	PP (mmHg)	0.33 ± 0.04	<0.0001	0.31 ± 0.04	<0.0001
	CCA media power	0.19 ± 0.03	<0.0001	0.22 ± 0.03	<0.0001
	T2DM (yes)	0.17 ± 0.04	0.0005		
	FPG (mmol/L)			0.12 ± 0.03	<0.005
	Cumulative R^2^	0.25	<0.0001	0.25	<0.0001

PP: pulse pressure, FPG: fasting plasma glucose.

## Data Availability

The data presented in this study are available on request from the corresponding author due to legal reasons.

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
