# Peer review of "Diabetes-Related Changes in Carotid Wall Properties: Role of Triglycerides"

_jcm, 2024, doi:10.3390/jcm13185654_

Round 1
Reviewer 1 Report
Comments and Suggestions for Authors
The present investigation aimed at demonstrating differences in carotid wall stiffness between T2DM patients vs. non-T2DM persons.
Although the topic and aims of the study appear quite interesting and a tremendous work was done by authors, numerous flaws in the methodology and presentation should be corrected before resubmission and publication.
They will be listed and explained in chronological order.
Introduction: The authors stated that the ultrasound probe receives radiofrequency signals. Although all electromagnetic waves between 30 KHz and 300 MHz are classified as radiofrequencies, in the case of ultrasound, these are mechanical waves, emitted and received by the transducer, which are then transformed by the piezoelectric effect into electric signals, popularly speaking. It seems that the authors have something confused in the understanding of the physical basis of ultrasound. It is not clear, why the authors decided to investigate the arterial stiffness, and particularly, the carotid wall stiffness. They stated that aortic stiffness is associated with diabetes, so why they chose the carotid wall? And what, in their opinion, is defined as stiffness? The link to the elasticity module or other physical parameters should be explained. The parameters they measured or calculated should be explained further. Particularly, why the authors believe that these parameters reflect the “stiffness” of the carotid wall.
M&M: No inclusion or exclusion criteria were defined. Were all patients consecutively included that entered the institution (would be rather few considering the long inclusion interval)? A chart flow of included or excluded patients would possibly demonstrate better how the patients became participants. Did they sign informed consent? Since the non-T2DM persons seemingly also entered the institution for diagnostics and treatment, it would be of interest for what reasons. Where these persons also patients suffering from other vascular entities or simply persons seeking vascular screening, but otherwise healthy?
The study design should be described (prospective, observational, randomized etc.??)
The ethical approval was in 2008, however, the inclusion of patients started only in 2015. The long interval should be explained. Local and European law may have substantially changed during that time, especially concerning data security, how was this taken into consideration? T2DM was defined as the elevation of fasting plasma glucose seemingly corresponding to WHO-criteria, however, the elevation should be confirmed twice to classify patients as suffering from diabetes following the definition. Obviously, this was not done. At which time of the day was the fasting glucose analysis done?
How was T2DM differentiated from other hyperglycemic conditions, such as T1DM or secondary DM?
Where the investigators blinded to the diagnosis of the patients?
Why did the investigators not follow-up the patients? There was a 100% chance to evaluate the predictive correctness of their values, and this would have substantially improved the value of the publication.
The 10 MHz linear Ultrasound probe may have sufficient resolution for clinical decisions and documentation, but for scientific applications, in particular consisting of distance measurements, a higher resolution would be desirable. Moreover, the used Device (Esaote MylabOne) is not very common in Europe, and the Image Gallery at the product website does not show a sufficient image quality or resolution to measure wall thickness. CEUS and Elastography are not supported by the device. It appears that the device has a focus on mobility, not on image quality, judging by the product photos.
Concerning the velocity measurements, the authors stated: Local carotid wave speed (m/s) was estimated from distension curves as previously described 21.
In fact, the device does not support PW doppler, as stated at the product website, and measuring a highly critical parameter, such as peak systolic velocity (or what else is meant by “wave speed”?) by estimations from mysterious curves seems to be an adventure rather than science.
It is not clear whether the measurements were done in longitudinal or axial projection. Figures would have been helpful in this regard.
A lot of indirect parameters are calculated by the measurements, i.e. WTS, by some formulas using a MATLAB package, which should by described in more detail. When only the listed formulas were used, this can also easily be done with a pocket calculator or an excel table. What was the reason to use this software? Moreover, it seems that the calculation is based on a cited publication (22) from 1999, which has a historical dimension considering the technical progress in ultrasound during the last 25 years. The device can only transfer data in the usual file formats (DICOM, Jpg, Mpg etc.), therefore, if raw data were transferred directly from the transducer to the MATLAB package, as the text suggests, this should be described in extensive detail.
The JMP software is not commonly used in medical statistics. Its obvious advantage is the ease of use for statistically not advanced users, but this is also the disadvantage, because it enables some “analysis” by mouse clickings without deeper understanding. While in creative software, like Photoshop or Word, this may suffice, medical statistics demand a more profound insight into the mathematical basis of the methods. Otherwise, false or questionable results are produced and may lead to erroneous assumptions and unsupported conclusions. In this case, a multiple regression analysis was performed without describing the details, such as the used parameters and algorithms (alpha, backward or forward regression and so on). As will later be shown, no senseful result is produced. The current version is JMP 18, the authors used 3.1, did they use a licensed product, if yes, why did they not use the current version?
There is also JMP clinical, which supports clinical trials. Why did the authors not use this module?
Results: Both cohorts differed in almost all covariates (table 2). Following table 4, most covariates were significantly associated with the response variables. Nevertheless, both cohorts were compared without matching (table 3). The p-value was seemingly adjusted for age, sex and BMI, but how was this done? A matched pairs analysis including all relevant covariates leaving the grouping variable as the only measured or defined unbalanced parameter, would be more appropriate, when the cohorts have not been randomized (which was obviously not the case). In the present analysis, relevant confounding bias is probable.
LDL-cholesterol was significantly higher in the non-T2DM cohort (see table 2). The opposite would be expected. The authors should explain this. Were patients with and without lipid lowering medications included?
Table 4 shows results from the multiple regression analysis, but does it show the initial or the final model, if the last, which other parameters had been included, how was the model reduced, which distribution family (normal, lognormal, poission etc.) was used for the response variables? The latter were probably not normally distributed because they were calculated values. It would have been better to use directly measured values, because these are more often normally distributed. Why did the authors include and exclude T2DM as a factor? In summary the table tells nothing about what it intends to tell.
Discussion: It remains unclear whether the measurements and the calculated parameters have anything to do with the carotid wall stiffness, because stiffness is a physical parameter (elasticity module?) which can be at best approximated. Elastography is one of the more or less established methods for this. Why did the authors not compare their methods with the established ones? The ROI was 1 cm proximal to the bifurcation, which means, that it may have interfered with the most typical predilection site for localized plaques. Did the authors exclude those patients with a higher grade stenosis (>50% NASCET) ? Moreover, these plaques may be hard or soft (in the carotids, the soft plaques seem to be more prone to embolization and cerebral ischemias), did the authors take this into consideration?
Considering the above mentioned issues in the methodology and analysis, none of the conclusions can be confirmed by the investigation. Moreover, the authors stated that:
“Since arterial stiffness is a strong predictor of CV and all-cause mortality…”. There is no evidence to support this thesis, otherwise the authors may cite corresponding publications, or even better, produce their own follow-up data.
In summary this manuscript necessitates a profound re-analysis and rewriting.
Author Response
Please, se attachment.

Reviewer 2 Report
Comments and Suggestions for Authors
I am grateful to the editor for the opportunity to review the manuscript by Michael Kozakova et al. "Diabetes-Related Changes in Carotid Wall Properties: Role of Triglycerides". In this article, the authors studied in detail the effect of diabetes on the properties of the arterial wall, especially taking into account the following parameters: media thickness, luminal diameter, wall tensile stress (WTS), wave speed and media power. The authors analyzed for the first time the radiofrequency signal reflected from the carotid artery wall in order to obtain information about changes in the carotid environment associated with diabetes. This allowed us to obtain some new scientific facts that may be useful for research in this area.
While reviewing, I had the following comments and questions:
1. The authors note that "The study population consisted of 540 individuals free of overt CV disease ..." (line 71). At the same time, 24% of the examined patients were taking Hypertension therapy (in the cohort as a whole), and 43% of patients in the group with diabetes mellitus. Does this mean that the authors did not consider the presence of arterial hypertension as a cardiovascular disease? If so, how correct is this statement? In addition, it is quite possible that some of the examined patients either did not know about their arterial hypertension, or knew about it but did not take Hypertension therapy. How did the authors of the article take these factors into account?
2. A large number of patients (15% among those examined without diabetes and 54% of patients with diabetes mellitus) were taking lipid-lowering therapy. There is a high probability that they have concomitant cardiovascular pathology. How did the authors take this factor into account?
3. The information presented in Table 1 is then presented in Table 2, already comparing the two groups. Apparently, Table 1 is redundant, and if the authors wish to present information on the entire cohort of patients, they could have simply introduced an additional column in Table 2.
3. The authors noted in the Statistical Analysis section that "Multiple regression analysis was used to identify the independent determinants of CCA media power and wave speed" (lines 130-132). This description is too sparse, it remains unclear which parameters the authors included in the regression analysis model and what method they used to select the parameters for this analysis.
4. The Conclusions section should not include references and information from other studies. This is more appropriately provided in the Discussion section (for example, when considering the clinical significance of this study).
Comments on the Quality of English LanguageNo comments
Round 2
Reviewer 2 Report
Comments and Suggestions for Authors
The authors responded to my comments and questions and made corrections to the text of the manuscript. I have no other comments.
Comments on the Quality of English LanguageNo comments